# The Parallel Structure–Activity Relationship Screening of Three Compounds Identifies the Common Agonist Pharmacophore of Pyrrolidine Bis-Cyclic Guanidine Melanocortin-3 Receptor (MC3R) Small-Molecule Ligands

**DOI:** 10.3390/ijms241210145

**Published:** 2023-06-14

**Authors:** Mark D. Ericson, Katie T. Freeman, Travis M. LaVoi, Haley M. Donow, Radleigh G. Santos, Marc A. Giulianotti, Clemencia Pinilla, Richard A. Houghten, Carrie Haskell-Luevano

**Affiliations:** 1Department of Medicinal Chemistry & Institute for Translational Neuroscience, University of Minnesota, Minneapolis, MN 55455, USA; erics063@umn.edu (M.D.E.); freem236@umn.edu (K.T.F.); 2Center for Translational Science, Florida International University, Port St. Lucie, FL 34987, USA; tlavoi@fiu.edu (T.M.L.); hdonow@fiu.edu (H.M.D.); mgiulian@fiu.edu (M.A.G.); cpinilla@fiu.edu (C.P.); houghten@tpims.org (R.A.H.); 3Department of Mathematics, Nova Southeastern University, Fort Lauderdale, FL 33314, USA; radleigh@nova.edu

**Keywords:** mMC3R small-molecule ligands, parallel structure–activity relationship studies, mMC5R fragments

## Abstract

The melanocortin receptors are involved in numerous physiological pathways, including appetite, skin and hair pigmentation, and steroidogenesis. In particular, the melanocortin-3 receptor (MC3R) is involved in fat storage, food intake, and energy homeostasis. Small-molecule ligands developed for the MC3R may serve as therapeutic lead compounds for treating disease states of energy disequilibrium. Herein, three previously reported pyrrolidine bis-cyclic guanidine compounds with five sites for molecular diversity (R1–R5) were subjected to parallel structure–activity relationship studies to identify the common pharmacophore of this scaffold series required for full agonism at the MC3R. The R2, R3, and R5 positions were required for full MC3R efficacy, while truncation of either the R1 or R4 positions in all three compounds resulted in full MC3R agonists. Two additional fragments, featuring molecular weights below 300 Da, were also identified that possessed full agonist efficacy and micromolar potencies at the mMC5R. These SAR experiments may be useful in generating new small-molecule ligands and chemical probes for the melanocortin receptors to help elucidate their roles in vivo and as therapeutic lead compounds.

## 1. Introduction

The melanocortin family consists of five known receptor subtypes (numbered 1–5) [1,2,3,4,5,6,7,8], endogenous agonists including α-MSH, β-MSH, γ-MSH, and ACTH derived from the proopiomelanocortin gene transcript [9], and the naturally occurring antagonists agouti and agouti-related protein (AGRP) [10,11,12,13,14,15]. This receptor family is involved in many biological functions that are linked to specific melanocortin receptor subtypes, such as pigmentation with the MC1R [2,6], steroidogenesis with the MC2R [6], and exocrine gland function in rodents with the MC5R [16]. The MC4R has been extensively studied for its role in food intake and appetite. Individuals with MC4R loss-of-function polymorphisms present with an obese phenotype [17,18], a phenotype that is also observed in MC4R KO mice [19]. The MC3R is similarly linked to energy homeostasis. Mice that lack the MC3R have an increased fat mass-to-lean mass ratio while maintaining a similar body weight to wild-type littermates [20,21]. Reduced food intake has also been reported for male MC3R KO mice maintained on a regular chow diet, with increased feed efficiency observed for both male and female MC3R KO mice [21]. The altered mass ratio has been observed in one individual homozygous for a purported MC3R loss-of-function polymorphism [22]. Although the MC3R has also been suggested as a potential target for anorexia, with an MC3R agonist ligand that has been claimed to result in an increased food intake in rodents [23], the dual nanomolar MC3R agonist/MC4R antagonist pharmacology of the Ac-Arg-Arg-DPhe(pI)-Tic-NH_2_ tetrapeptide [24] utilized complicates the interpretation of the receptors responsible for the observed increased feeding response as it is well recognized that MC4R antagonist ligands increase food intake responses [25]. While the central administration of melanocortin agonists is known to decrease food intake in wild-type mice [25,26,27], MC4R KO mice (with intact MC3R) exhibited a dose-dependent decrease in food intake when the melanocortin agonist Ac-His-DPhe-Arg-Trp-NH_2_ was administered, suggesting that the activation of the MC3R decreases food intake [28]. Due to these conflicting reports, the development of MC3R ligands may help clarify the role of the MC3R on food intake and be useful in the development of therapeutic lead compounds for disease states of energy disorder.

To identify potential MC3R-selective small molecules, 69 distinct mixture-based scaffold libraries were previously assayed at the MC3R and MC4R [29]. Using a combined metric of MC3R activity as well as MC3R over MC4R selectivity, both pyrrolidine bis-cyclic guanidine and pentaamine scaffolds were advanced to a mixture-based positional scan [29], an approach previously utilized to identify MC4R polymorphic rescue compounds [30], MC3R agonist tetrapeptides [24,31], and MC4R antagonist ligands [32]. While many of the pentaamine mixtures and individual compounds were observed to be toxic, deconvolution of the pyrrolidine bis-cyclic guanidine library led to the synthesis of 37 compounds, of which 9 possessed full agonist efficacy at the MC3R with sub-micromolar potencies [29]. While similar agonist efficacies and potencies were observed for these 9 compounds at the MC1R and MC5R, none were full agonists at the MC4R and all 9 possessed antagonist activity at the MC4R (pA_2_ values between 5.5 and 7.0) [29]. Due to the observed 10-fold selectivity for the MC3R over MC4R, performing additional structure–activity relationship (SAR) studies on pyrrolidine bis-cyclic guanidine compounds will help identify the key pharmacophore of this scaffold series, and may lead to more potent, selective ligands that can be used as lead compounds in probing the in vivo physiological functions of the MC3R.

The identified pyrrolidine bis-cyclic guanidine scaffold was synthesized as a resin-bound tetrapeptide with a variety of capping carboxylic acids [29]. Following reduction of the amide carbonyl moieties using a borane-tetrahydrofuran complex (which does not alter the α-carbon stereochemistry) [33,34], the amine backbone was cyclized on-resin with cyanogen bromide before global deprotection and cleavage from the polystyrene support with anhydrous HF (Figure 1). Since the precursor to the final product is a peptide, multiple compounds can be synthesized simultaneously, permitting the rapid synthesis of a number of analogs to varying at select positions, utilizing the “tea-bag” method for solid-phase synthesis [35]. Due to the ease of parallel synthesis, SAR around multiple starting ligands can be readily explored. Previously, parallel functional alanine scans of the endogenous α-MSH and synthetic NDP-MSH tridecapeptide melanocortin agonists, which differ by two amino acids, indicated that altered potencies due to substitutions in α-MSH did not necessarily correlate with equivalent substitutions in NDP-MSH [36]. These experiments suggest that structural modifications in one melanocortin ligand may not be predictive for similar compounds. Performing SAR studies across multiple templates may identify unique positions of interest in different compounds, as well as the core pharmacophore that is important for activity across a given scaffold series.

Herein, three previously identified pyrrolidine bis-cyclic guanidine compounds were selected as leads for initial SAR studies [29]. For each of the starting compounds, a series of truncated and stereochemical analogs were made to explore the importance of the five substitution positions (R1–R5). All compounds were screened for agonist activity at the MC1R, MC3R, MC4R, and MC5R. Since all compounds with full MC3R efficacy did not possess full MC4R efficacy, these ligands were also assayed for antagonist activity at the MC4R. Synthesizing and screening the compound series in parallel allowed for the identification of the common pharmacophore unit correlating with full agonist efficacy at the MC3R, and to examine if SAR trends in one ligand were present in the other compounds.

## 2. Results

Three compounds (2718.001, 2718.002, and 2718.003; Figure 1) were selected from a library of previously reported pyrrolidine bis-cyclic guanidine compounds (corresponding to the prior publication compounds numbered **11**, **2**, and **23**, respectively [29]). Whereas all of the previously reported compounds possessed a Pro to generate the pyrrolidine functionality [29], the stereochemistry of this group was varied in the present study, introducing a new substitution point. While the R1 functionality remains constant between the prior publication and the experiments reported herein, the R2 position herein represents the stereochemistry of the pyrrolidine group. Therefore, the prior publication groups R2, R3, and R4 are reported herein as R3, R4, and R5, respectively. Compounds 2718.001 and 2718.002 both possess R1 *R*-isobutyl, R2 *S*-pyrrolidine, R4 (*S*,*S*)-1-hydroxyethyl, R5 4-*t*butyl-cyclohexyl-methyl functionalities, and vary at the R3 position (*R*-benzyl in 2718.001 and *R*-cyclohexyl-methyl in 2718.002; Figure 1). Both compounds were reported to possess similar mMC3R agonist potency (EC_50_ = 310 nM for 2718.001 and 210 nM for 2718.002) [29]. Performing parallel SAR studies on both ligands probes the importance of the R3 position for activity, as well as if the R3 position alters the minimal pharmacophore necessary for full MC3R agonist activity. Altering the R1 position from *R*-isobutyl to *R*-isopropyl while maintaining the same R2, R3, R4, and R5 groups as 2718.001 and 2718.002 identified additional compounds with similar mMC3R potencies [29]. Whereas compounds with an R1 *R*-isobutyl group required an R4 (*S*,*S*)-1-hydroxyethyl for observable mMC3R agonist potency, several compounds with an R1 *R*-isopropyl group and an R4 *R*-propyl or *R*-isopropyl possessed full agonist efficacy at the mMC3R [29]. The most potent of these compounds (2718.003 herein) possessed an R1 *R*-isopropyl, R2 *S*-pyrrolidine, R3 *R*-cyclohexyl-methyl, R4 *R*-propyl, and R5 4-*t*butyl-cyclohexyl-methyl (mMC3R EC_50_ = 350 nM) [29]. Due to the incorporated R1 and R4 changes, this compound was selected as another starting compound to determine the importance of these substitutions for MC3R activity (Figure 1).

Several parallel modifications to the three compounds were performed to identify SAR trends. Functional groups were serially truncated off each compound to identify the importance of each substitution point, as well as generating R4 and R5 fragments for each scaffold. The stereochemistry at individual positions was also sequentially varied to identify any potential effects of altering chirality. The enantiomer form of each starting compound (where all stereochemical centers were inverted) was also generated to examine the effects of globally altering the stereochemistry. Specifically for 2718.001, analogs consisting of two and three positions with inverted stereocenters were generated. Due to the reported similar mMC3R potencies of 2718.001 and 2718.002, varying only at the R3 position (the aromatic *R*-benzyl and aliphatic *R*-cyclohexyl-methyl groups, respectively), a set of compounds was generated that varied aromatic (*R*-4-hydroxy-benzyl, *R*-(1*H*-indol-3-yl)-methyl, and *R*-2-methyl-naphthalene) and aliphatic (*R*-methyl, *R*-butyl, and *R*-cyclohexyl) substitutions at the R3 position while maintaining R1 *R*-isobutyl, R2 *S*-pyrrolidine, R4 (*S*,*S*)-1-hydroxyethyl, and R5 4-*t*butyl-cyclohexyl-methyl functionalities.

Individual pyrrolidine bis-cyclic guanidine compounds were synthesized on a solid support using N-*α*-Boc compatible chemistry (Figure 1), as previously described [29]. Traditional solid-phase peptide synthesis generated capped peptides in parallel. While on the solid support, the backbone amides were transformed into the corresponding amines by an established borane reduction of the carbonyl without epimerization of the sidechain α-carbon [29,33,34]. Following the reduction step, cyclization of the polyamine functionalities with cyanogen bromide yielded the desired pyrrolidine bis-cyclic guanidine compounds. The final products were globally sidechain deprotected and cleaved off the polystyrene resin support with anhydrous hydrogen fluoride (HF). After removal of the excess HF, the individual compounds were purified using HPLC. Of the 52 synthesized compounds, 50 were purified to >95% purity, as determined by peak integration at 214 nM. One compound (2718.002) was purified to >94% purity, and one compound (2718.024) was purified to >84% purity. The compounds were characterized by ^1^H NMR and LC/MS (Appendix A).

Pyrrolidine bis-cyclic guanidine compounds were assessed for biological activity using the AlphaScreen cAMP kit with HEK293 cells stably expressing the mMC1R, mMC3R, mMC4R, or mMC5R, as previously described [37,38,39]. Since the MC2R is only stimulated by ACTH and no other endogenously derived POMC agonists [40], it was not examined in the current study. The synthetic melanocortin agonist peptide NDP-MSH was included as a control compound. Compounds were assayed for agonist activity in a 7-point dose–response curve (10^−3^ to 10^−9^ M). Ligands that possessed full mMC3R agonist efficacy (relative to NDP-MSH) and partially activated the mMC4R were assayed for antagonist activity using the synthetic melanocortin agonist NDP-MSH in a Schild experimental paradigm [41]. Pyrrolidine bis-cyclic guanidines that did not possess agonist activity in two independent experiments were considered inactive at the concentrations assayed and were not further studied. Active compounds were assayed in at least three independent experiments. Compounds that possessed sigmoidal dose–response curves and partial agonist efficacy (<90% NDP-MSH signal) were binned into two groups (A = 20–50% NDP-MSH, B = 51–90% NDP-MSH; Figure 2). Due to the inherent experimental error associated with these types of assays, we considered compounds that were at within a 3-fold potency range to be equipotent. Since the AlphaScreen cAMP assay is a loss-of-signal assay, in which higher concentrations of compound result in lower assay signal, the data were normalized to baseline and maximal NDP-MSH signal for illustrative purposes, as previously described [38,39].

### 2.1. Structure–Activity Relationship Studies of 2718.001

The first lead compound (2718.001) was previously reported to possess full mMC3R and mMC5R agonist activity (EC_50_ = 310 and 530 nM, respectively), partially stimulated the mMC1R (58% efficacy at 10 µM concentrations) and the mMC4R (10–50% activation at 10 µM concentrations), and was an antagonist at the mMC4R (pA_2_ = 7.0 with agonist MTII) [29] when assayed in a β-galactosidase reporter gene assay [42]. In the present experiment (assayed with the cAMP AlphaScreen kit), full agonist efficacy was observed at the mMC3R and mMC5R (EC_50_ = 270 and 260 nM, respectively; Table 1 and Figure 3). Full agonist activity was also observed for the mMC1R (EC_50_ = 320 nM; Figure 3), unlike the partial response previously observed. This difference may be due to the different assays used to examine activity. Partial activation of the mMC4R was observed in the present study (60% at 1 mM concentrations), and 2718.001 possessed antagonist activity at the mMC4R (pA_2_ = 6.0 with agonist NDP-MSH; Figure 4), supporting the MC3R agonist/MC4R partial response and antagonist response of 2718.001 previously reported.
ijms-24-10145-t001_Table 1Table 1Agonist Pharmacology of Lead Molecule 2718.001 Analogs ^a^.Compound IDChemical StructureFunctionalitiesmMC1RmMC3RmMC4RmMC5REC_50_ (nM)EC_50_ (nM)EC_50_ (nM)pA_2_EC_50_ (nM)NDP-MSHAc-Ser-Tyr-Ser-Nle-Glu-His-DPhe-Arg-Trp-Gly-Lys-Pro-Val-NH_2_0.055 ± 0.0080.069 ± 0.0090.31 ± 0.04
0.10 ± 0.012718.001
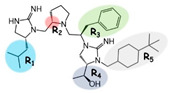
R1: *R*-isobutylR2: *S*-pyrrolidineR3: *R*-benzylR4: (*S*,*S*)-1-hydroxyethylR5: 4-*t*butyl-cyclohexyl-methyl320 ± 30270 ± 2060% @ 1 mM6.0 ± 0.1260 ± 30**Functionality Truncation Analogs**2718.024
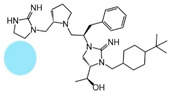
R1: **hydrogen**R2: *S*-pyrrolidineR3: *R*-benzylR4: (*S*,*S*)-1-hydroxyethylR5: 4-*t*butyl-cyclohexyl-methyl390 ± 503000 ± 70050% @ 1 mM6.3 ± 0.1480 ± 502718.023
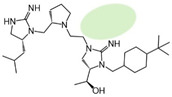
R1: *R*-isobutylR2: *S*-pyrrolidineR3: **hydrogen**R4: (*S*,*S*)-1-hydroxyethylR5: 4-*t*butyl-cyclohexyl-methyl85% @ 1 mM30% @ 1 mM40,000 ± 20,000 (A)
80% @ 1 mM2718.022
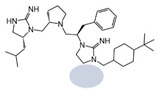
R1: *R*-isobutylR2: *S*-pyrrolidineR3: *R*-benzylR4: **hydrogen**R5: 4-*t*butyl-cyclohexyl-methyl400 ± 40380 ± 7050% @ 1 mM6.1 ± 0.1320 ± 102718.004
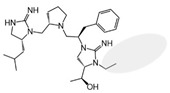
R1: *R*-isobutylR2: *S*-pyrrolidineR3: *R*-benzylR4: (*S*,*S*)-1-hydroxyethylR5: **ethyl**20,600 ± 600 (A)>1,000,000>1,000,000
>1,000,0002718.006
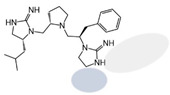
R1: *R*-isobutylR2: *S*-pyrrolidineR3: *R*-benzylR4: **hydrogen**R5: **hydrogen**40% @ 1 mM>1,000,00020% @ 1 mM
30% @ 1 mM2718.020
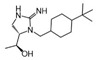
R1: R2: R3: R4: (*S*,*S*)-1-hydroxyethylR5: 4-*t*butyl-cyclohexyl-methyl70% @ 1 mM70% @ 1 mM55% @ 1 mM
140,000 ± 30,000**Single Residue Stereocenter Inversion Analogs**2718.033
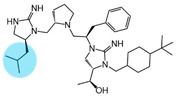
R1: ***S*-isobutyl**R2: *S*-pyrrolidineR3: *R*-benzylR4: (*S*,*S*)-1-hydroxyethylR5: 4-*t*butyl-cyclohexyl-methyl380 ± 403800 ± 600 (B)60% @ 1 mM
900 ± 3002718.032
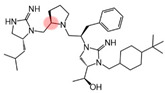
R1: *R*-isobutylR2: ***R*-pyrrolidine**R3: *R*-benzylR4: (*S*,*S*)-1-hydroxyethylR5: 4-*t*butyl-cyclohexyl-methyl400 ± 601500 ± 400 (B)55% @ 1 mM
600 ± 2002718.031
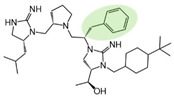
R1: *R*-isobutylR2: *S*-pyrrolidineR3: *S*-**benzyl**R4: (*S*,*S*)-1-hydroxyethylR5: 4-*t*butyl-cyclohexyl-methyl2300 ± 90055% @ 1 mM55% @ 1 mM
70% @ 1 mM2718.030
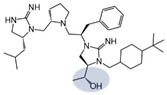
R1: *R*-isobutylR2: *S*-pyrrolidineR3: *R*-benzylR4: **(*R*,*R*)-1-hydroxyethyl**R5: 4-*t*butyl-cyclohexyl-methyl4000 ± 2000 (B)50% @ 1 mM60% @ 1 mM
5000 ± 2000 (B)**Double Residue Stereocenter Inversion Analogs**2718.040
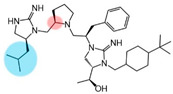
R1: ***S*-isobutyl**R2: ***R*-pyrrolidine**R3: *R*-benzylR4: (*S*,*S*)-1-hydroxyethylR5: 4-*t*butyl-cyclohexyl-methyl900 ± 3003700 ± 400 (B)60% @ 1 mM
500 ± 302718.039
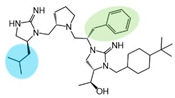
R1: ***S*-isobutyl**R2: *S*-pyrrolidineR3: ***S*-benzyl**R4: (*S*,*S*)-1-hydroxyethylR5: 4-*t*butyl-cyclohexyl-methyl4000 ± 2000(B)50% @ 1 mM55% @ 1 mM
45,000 ± 5000 (B)2718.037
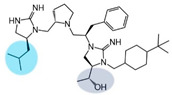
R1: ***S*-isobutyl**R2: *S*-pyrrolidineR3: *R*-benzylR4: **(*R*,*R*)-1-hydroxyethyl**R5: 4-*t*butyl-cyclohexyl-methyl900 ± 200 (B)50% @ 1 mM55% @ 1 mM
7000 ± 1000 (B)2718.038
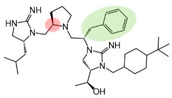
R1: *R*-isobutylR2: ***R*-pyrrolidine**R3: ***S*-benzyl**R4: (*S*,*S*)-1-hydroxyethylR5: 4-*t*butyl-cyclohexyl-methyl600 ± 1005900 ± 900 (B)1600 ± 300 (B)
1400 ± 3002718.036
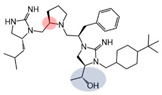
R1: *R*-isobutylR2: ***R*-pyrrolidine**R3: *R*-benzylR4: **(*R*,*R*)-1-hydroxyethyl**R5: 4-*t*butyl-cyclohexyl-methyl900 ± 30050% @ 1 mM55% @ 1 mM
5800 ± 900 (B)2718.035
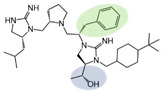
R1: *R*-isobutylR2: *S*-pyrrolidineR3: ***S*-benzyl**R4: **(*R*,*R*)-1-hydroxyethyl**R5: 4-*t*butyl-cyclohexyl-methyl3700 ± 70055% @ 1 mM55% @ 1 mM
23,000 ± 9000 (B)**Triple Residue Stereocenter Inversion Analogs**2718.044
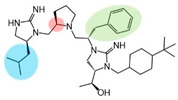
R1: ***S*-isobutyl**R2: ***R*-pyrrolidine**R3: ***S*-benzyl**R4: (*S*,*S*)-1-hydroxyethylR5: 4-*t*butyl-cyclohexyl-methyl7000 ± 2000 (B)5600 ± 800 (A)50% @ 1 mM
2900 ± 900 (B)2718.043
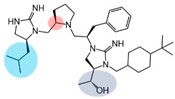
R1: ***S*-isobutyl**R2: ***R*-pyrrolidine**R3: *R*-benzylR4: **(*R*,*R*)-1-hydroxyethyl**R5: 4-*t*butyl-cyclohexyl-methyl3500 ± 200 (B)55% @ 1 mM55% @ 1 mM
8000 ± 2000 (B)2718.042
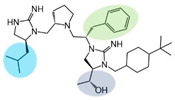
R1: ***S*-isobutyl**R2: *S*-pyrrolidineR3: ***S*-benzyl**R4: **(*R*,*R*)-1-hydroxyethyl**R5: 4-*t*butyl-cyclohexyl-methyl2900 ± 20065% @ 1 mM65% @ 1 mM
21,300 ± 700 (B)2718.041
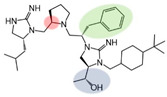
R1: *R*-isobutylR2: ***R*-pyrrolidine**R3: ***S*-benzyl**R4: **(*R*,*R*)-1-hydroxyethyl**R5: 4-*t*butyl-cyclohexyl-methyl40,000 ± 10,00060% @ 1 mM60% @ 1 mM
8000 ± 1000 (B)**Quadruple Residue Stereocenter Inversion Analogs**2718.034
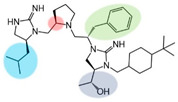
R1: ***S*-isobutyl**R2: ***R*-pyrrolidine**R3: ***S*-benzyl**R4: **(*R*,*R*)-1-hydroxyethyl**R5: 4-*t*butyl-cyclohexyl-methyl75% @ 1 mM50% @ 1 mM55% @ 1 mM
3000 ± 1000^a^ Pyrrolidine bis-cyclic guanidine compounds were assayed at the selected receptor subtypes for agonist activity from 10^−3^ to 10^−9^ M. The results are tabulated as the mean EC_50_ or pA_2_ values from at least three independent experiments with the reported error being the standard error of the mean (SEM). A percentage denotes the percent maximal stimulatory response (compared to NDP-MSH) observed at 1 mM concentrations, but not enough stimulation was observed to determine an EC_50_ value. The use of >1,000,000 indicates that the compound was examined but lacked agonist activity at concentrations up to 1 mM in at least two independent experiments. For partial agonist compounds (a sigmoidal dose–response curve was observed, but efficacy was less than 90% maximal NDP-MSH signal), compounds were binned into two groups: A (20–50% receptor activation) or B (51–90% receptor activation), Figure 2. For compounds possessing full agonist efficacy at the MC3R and that did not fully activate the MC4R, follow-up antagonist experiments and corresponding pA_2_ values, via a Schild analysis [41], were performed at the mMC4R.


Replacing the R4 (*S*,*S*)-1-hydroxyethyl with hydrogen (2718.022) resulted in equipotent agonist activity at the mMC1R (EC_50_ = 400 nM), mMC3R (EC_50_ = 380 nM), and mMC5R (EC_50_ = 320 nM) compared to 2718.001, with similar partial activation (50% at 1 mM concentrations) and antagonist activity (pA_2_ = 6.1) at the mMC4R. Similar agonist potency at the mMC1R (EC_50_ = 390 nM) and mMC5R (EC_50_ = 480 nM) and mMC4R antagonist activity (pA_2_ = 6.3) were also observed when replacing the R1 *R*-isobutyl group with hydrogen (2718.024) compared to 2718.001, although an 11-fold decrease in mMC3R potency (EC_50_ = 3000 nM) was also observed for this substitution. Truncation of the R3 position to hydrogen (2718.023) or the R5 position to ethyl (2718.004) did not result in compounds with full agonist efficacy, although 2718.023 induced a partial agonist response at the mMC4R (EC_50_ = 40,000 nM) and 2718.004 did result in partial agonism at the mMC1R (EC_50_ = 20,600; Figure 3). Replacing both the R4 and R5 positions with hydrogen (2718.006) resulted in minimal-to-no activity at 1 mM concentrations at all of the receptors, while incorporation of the cyclic guanidine containing the R4 (*S*,*S*)-1-hydroxyethyl and R5 4-*t*butyl-cyclohexyl-methyl groups resulted in full mMC5R agonist activity (EC_50_ = 140,000 nM; Figure 3).

Inverting single stereocenters of 2718.001 resulted in two compounds (2718.033 possessing an R1 *S*-isobutyl group and 2718.031 possessing an R2 *R*-pyrrolidine group) that possessed equipotent mMC1R and mMC5R agonist potencies relative to 2718.001. These two compounds also possessed partial agonist efficacies and micromolar potencies at the mMC3R, and partially activated the mMC4R. Inverting the R3 position to *S*-benzyl (2718.031) decreased mMC1R agonist potency 7-fold relative to 2718.001, and partially activated the mMC3R, mMC4R, and mMC5R. Changing the stereochemistry at the R4 position to (*R*,*R*)-1-hydroxyethyl resulted in a compound that was a partial agonist at the mMC1R (EC_50_ = 4000 nM) and mMC5R (EC_50_ = 5000 nM), while partially activating the mMC3R and mMC4R.

Simultaneously inverting the stereochemistry of two positions resulted in three equipotent mMC1R agonist compounds (2718.040 altering R1 and R2; 2719.038 altering R2 and R3, Figure 3; 2718.036 altering R2 and R4) relative to 2718.001. These double substitutions are the three double substitutions that invert the stereochemistry at the R2 position. Altering the R3 and R4 stereochemistry (2718.035) decreased mMC1R potency 12-fold relative to 2718.001, while partial agonist efficacy was observed for altering the R1 and R3 (2718.039) and R1 and R4 (2718.037) positions. None of the double stereochemical inversion compounds resulted in full agonist efficacy at the mMC3R or mMC4R. Two compounds (2718.038 and 2718.040) were full agonists at the mMC5R. The R1- and R2-inverted 2718.040 was an equipotent mMC5R agonist relative to 2718.001, while the 2718.038 (inverting the R2 and R3 positions) possessed 5-fold decreased potency at the mMC5R (Figure 3) relative to 2718.001.

Concurrent inversion of three stereocenters resulted in two compounds that possessed full mMC1R agonist efficacy. Inverting the R1, R3, and R4 positions (2718.042) and R2, R3, and R4 positions (2718.041) resulted in compounds with 9- and 125-fold decreased mMC1R potencies compared to 2718.001, while inverting the R1, R2, and R3 (2718.044) and R1, R2, and R4 (2718.043) resulted in partial mMC1R agonists with micromolar potencies. The triple stereochemical inversion compounds did not possess full mMC3R or mMC4R agonist efficacy and were partial agonists at the mMC5R. Inverting all four stereocenters (2718.034) resulted in partial activation of the mMC1R, mMC3R, and mMC4R at 1 mM concentrations. This compound was a full mMC5R agonist (EC_50_ = 3000 nM), with 12-fold decreased potency relative to 2718.001.

### 2.2. Structure–Activity Relationship Studies of 2718.002

The second compound examined, 2718.002, was previously reported to possess full mMC1R, mMC3R, and mMC5R agonist activity (EC_50_ = 300, 210, and 140 nM, respectively), partial agonist efficacy at the mMC4R (45% MTII signal, EC_50_ = 270 nM), and antagonist activity at the mMC4R (pA_2_ = 5.8 with MTII agonist) [29]. The results are similar to the present report (Table 2) where full agonist activity was observed at the mMC1R (EC_50_ = 270 nM), mMC3R (EC_50_ = 240 nM), and mMC5R (EC_50_ = 290 nM). A partial agonist response was observed at the mMC4R (EC_50_ = 1200 nM), and 2718.002 possessed antagonist activity in the presence of NDP-MSH (pA_2_ = 6.2 with NDP-MSH agonist). These data support the observed mMC3R agonist and mMC4R partial agonist/antagonist pharmacology previously reported.
ijms-24-10145-t002_Table 2Table 2Agonist Pharmacology of Lead Molecule 2718.002 Analogs ^a^.Compound IDChemical StructureFunctionalitiesmMC1RmMC3RmMC4RmMC5REC_50_ (nM)EC_50_ (nM)EC_50_ (nM)pA_2_EC_50_ (nM)NDP-MSHAc-Ser-Tyr-Ser-Nle-Glu-His-DPhe-Arg-Trp-Gly-Lys-Pro-Val-NH_2_0.055 ± 0.0080.069 ± 0.0090.31 ± 0.04
0.10 ± 0.012718.002
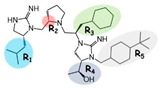
R1: *R*-isobutylR2: *S*-pyrrolidineR3: *R*-cyclohexyl-methylR4: (*S*,*S*)-1-hydroxyethylR5: 4-*t*butyl-cyclohexyl-methyl270 ± 20240 ± 81200 ± 300 (A)6.2 ± 0.1290 ± 10**Functionality Truncation Analogs**2718.026
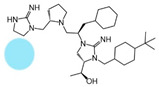
R1: **hydrogen**R2: *S*-pyrrolidineR3: *R*-cyclohexyl-methylR4: (*S*,*S*)-1-hydroxyethylR5: 4-*t*butyl-cyclohexyl-methyl460 ± 302600 ± 2001800 ± 500(A)6.2 ± 0.1300 ± 102718.025
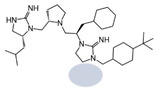
R1: *R*-isobutylR2: *S*-pyrrolidineR3: *R*-cyclohexyl-methylR4: **hydrogen**R5: 4-*t*butyl-cyclohexyl-methyl510 ± 901000 ± 30060% @ 1 mM6.1 ± 0.1600 ± 1002718.010
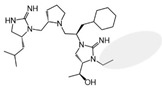
R1: *R*-isobutylR2: *S*-pyrrolidineR3: *R*-cyclohexyl-methylR4: (*S*,*S*)-1-hydroxyethylR5: **ethyl**5000 ± 1000(B)>1,000,00020% @ 1 mM
45,5000 ± 35,000 (A)2718.011
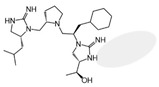
R1: *R*-isobutylR2: *S*-pyrrolidineR3: *R*-cyclohexyl-methylR4: (*S*,*S*)-1-hydroxyethylR5: **hydrogen**15,000 ± 4000(B)>1,000,000>1,000,000
40% @ 1 mM2718.012
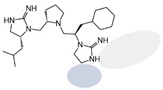
R1: *R*-isobutylR2: *S*-pyrrolidineR3: *R*-cyclohexyl-methylR4: **hydrogen**R5: **hydrogen**15,000 ± 4000(B)>1,000,000>1,000,000
25% @ 1 mM**Single Residue Stereocenter Inversion Analogs**2718.048
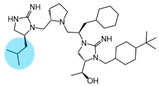
R1: ***S*-isobutyl**R2: *S*-pyrrolidineR3: *R*-cyclohexyl-methylR4: (*S*,*S*)-1-hydroxyethylR5: 4-*t*butyl-cyclohexyl-methyl390 ± 302900 ± 200(B)65% @ 1 mM
650 ± 802718.047
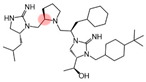
R1: *R*-isobutylR2: ***R*-pyrrolidine**R3: *R*-cyclohexyl-methylR4: (*S*,*S*)-1-hydroxyethylR5: 4-*t*butyl-cyclohexyl-methyl870 ± 7085% @ 10 µM3100 ± 200(B)
450 ± 302718.046
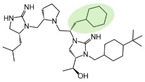
R1: *R*-isobutylR2: *S*-pyrrolidineR3: ***S*-cyclohexyl-methyl**R4: (*S*,*S*)-1-hydroxyethylR5: 4-*t*butyl-cyclohexyl-methyl1800 ± 2009000 ± 1000(B)55% @ 1 mM
5000 ± 1000**2718.045**
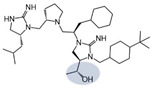
R1: *R*-isobutylR2: *S*-pyrrolidineR3: *R*-cyclohexyl-methylR4: **(*R*,*R*)-1-hydroxyethyl**R5: 4-*t*butyl-cyclohexyl-methyl1000 ± 1003500 ± 200(B)50% @ 1 mM
2200 ± 700**Quadruple Residue Stereocenter Inversion Analogs**2718.049
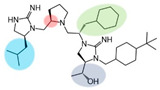
R1: ***S*-isobutyl**R2: ***R*-pyrrolidine**R3: ***S*-cyclohexyl-methyl**R4: **(*R*,*R*)-1-hydroxyethyl**R5: 4-*t*butyl-cyclohexyl-methyl8000 ± 2000 (B)65% @ 1 mM65% @ 1 mM
2800 ± 100^a^ Pyrrolidine bis-cyclic guanidine compounds were assayed at the selected receptor subtypes for agonist activity from 10^−3^ to 10^−9^ M. The results are tabulated as the mean EC_50_ or pA_2_ values from at least three independent experiments with the reported error being the standard error of the mean (SEM). A percentage denotes the percent maximal stimulatory response (compared to NDP-MSH) observed at 1 mM concentrations, but not enough stimulation was observed to determine an EC_50_ value. The use of >1,000,000 indicates that the compound was examined but lacked agonist activity at concentrations up to 1 mM in at least two independent experiments. For partial agonist compounds (a sigmoidal dose–response curve was observed, but efficacy was less than 90% maximal NDP-MSH signal), compounds were binned into two groups: A (20–50% receptor activation) or B (51–90% receptor activation), Figure 2. For compounds possessing full agonist efficacy at the MC3R and that did not fully activate the MC4R, follow-up antagonist experiments and corresponding pA_2_ values, via a Schild analysis [41], were performed at the mMC4R.


Replacing the R4 (*S*,*S*)-1-hydroxylethyl group with hydrogen (2718.025) decreased mMC3R agonist potency 4-fold relative to 2718.002, possessed equipotent mMC1R (EC_50_ = 510 nM) and mMC5R (EC_50_ = 600 nM) agonist activity, and was an equipotent mMC4R antagonist (pA_2_ = 6.1). Truncation of the R1 *R*-isobutyl group to hydrogen (2718.026) decreased mMC3R potency 11-fold relative to 2718.002. This compound was equipotent to 2718.002 at the mMC1R and mMC5R, and possessed similar mMC4R partial agonist (EC_50_ = 1800 nM) and antagonist (pA_2_ = 6.2 with NDP-MSH agonist) activities. Since 2718.001 and 2718.002 share R1, R2, R4, and R5 substitutions, the truncation of the R3 group (2718.023) was previously incorporated into Table 1. Truncation of the R5 4-*t*butyl-cyclohexyl-methyl group to ethyl (2718.010) or hydrogen (2718.011) resulted in partial agonist activity at the mMC1R, no recorded activity at the mMC3R up to 1 mM concentrations, minimal-to-no activation of the mM4R, and partial agonism (EC_50_ = 45,500 nM) and partial activation (40% at 1 mM concentrations) at the mMC5R, respectively. Simultaneous truncation of the R4 and R5 positions to hydrogen (2718.012) resulted in a partial agonist at the mMC1R (EC_50_ = 15,000 nM), no stimulation of the mMC3R and mMC4R, and partial activation of the mMC5R (25%) at the highest concentrations assayed.

Inverting the stereochemistry of the R1 position in this scaffold to *S*-isobutyl (2718.048) resulted in an equipotent mMC1R (EC_50_ = 390 nM) and mMC5R (EC_50_ = 650 nM) agonist compared to 2718.002. This compound was a partial agonist at the mMC3R (EC_50_ = 2900 nM) and partially activated the mMC4R (65%) at 1 mM concentrations. Changing the R2 stereochemistry to *R*-pyrrolidine (2718.047) also maintained mMC1R and mMC5R potency relative to 2718.002. This compound did not fully stimulate the mMC3R and was a partial agonist at the mMC4R (EC_50_ = 3100 nM). Changing the R3 position to *S*-cyclohexyl-methyl (2718.046) or the R4 position to *R*,*R*-1-hydroxyethyl (2718.045) resulted in decreased mMC1R (7- and 4-fold, respectively) and mMC5R (17- and 8-fold, respectively) agonist potency compared to 2718.002. Both compounds were partial agonists at the mMC3R, and partially activated the mMC4R at 1 mM concentrations. While this series did not examine double or triple stereochemical inversions, changing the stereochemistry at the R1, R2, R3, and R4 positions (2718.049) resulted in a partial agonist at the mMC1R (EC_50_ = 8000 nM), partial activation of the mMC3R and mMC4R (65% at 1 mM concentrations), and was a full agonist at the mMC5R (EC_50_ = 2800 nM), possessing 10-fold decreased potency relative to 2718.002.

### 2.3. Structure–Activity Relationship Studies at the R3 Position of 2718.001 and 2718.002

The 2718.001 and 2718.002 lead compounds both possess an R1 *R*-isobutyl, R2 *S*-pyrrolidine, R4 (*S*,*S*)-1-hydroxylethyl, and R5 4-*t*butyl-cyclohexyl-methyl groups. The 2718.001 bis-cyclic guanidine possesses an R3 *R*-benzyl, while 2718.002 has an R3 *R*-cyclohexyl-methyl functionality. A set of six substitutions (Table 3), comprised of aliphatic and aromatic substitutions, were incorporated into the R3 position. Both 2818.001 and 2718.002 possess similar mMC1R (EC_50_ = 320 and 270 nM, respectively), mMC3R (EC_50_ = 270 and 240 nM, respectively), and mMC5R (EC_50_ = 260 and 290 nM, respectively) agonist potencies. The R3-substituted compounds are compared to 2718.002 herein, but can be compared to 2718.001 with similar relative activities.
ijms-24-10145-t003_Table 3Table 3Agonist Pharmacology of 2718.001 and 2718.002 Analogs Substituted at the R3 Position ^a^.Compound IDChemical StructureFunctionalitiesmMC1RmMC3RmMC4RmMC5REC_50_ (nM)EC_50_ (nM)EC_50_ (nM)EC_50_ (nM)NDP-MSHAc-Ser-Tyr-Ser-Nle-Glu-His-DPhe-Arg-Trp-Gly-Lys-Pro-Val-NH_2_0.055 ± 0.0080.069 ± 0.0090.31 ± 0.040.10 ± 0.01**R3 Substitution Analogs**2718.061
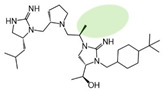
R1: *R*-isobutylR2: *S*-pyrrolidineR3: ***R*-methyl**R4: (*S*,*S*)-1-hydroxyethylR5: 4-*t*butyl-cyclohexyl-methyl65% @ 1 mM20% @ 1 mM>1,000,00065% @ 1 mM2718.060
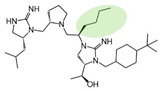
R1: *R*-isobutylR2: *S*-pyrrolidineR3: ***R*-butyl**R4: (*S*,*S*)-1-hydroxyethylR5: 4-*t*butyl-cyclohexyl-methyl2200 ± 700(B)2200 ± 200(B)45% @ 1 mM1100 ± 3002718.057
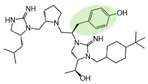
R1: *R*-isobutylR2: *S*-pyrrolidineR3: ***R*-4-hydroxy-benzyl**R4: (*S*,*S*)-1-hydroxyethylR5: 4-*t*butyl-cyclohexyl-methyl500 ± 10011,000 ± 2000(B)50% @ 1 mM6000 ± 2000(B)2718.055
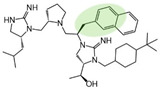
R1: *R*-isobutylR2: *S*-pyrrolidineR3: ***R*-2-methyl-naphthalene**R4: (*S*,*S*)-1-hydroxyethylR5: 4-*t*butyl-cyclohexyl-methyl1000 ± 2007000 ± 3000(B)70% @ 1 mM8000 ± 20002718.056
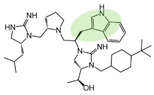
R1: *R*-isobutylR2: *S*-pyrrolidineR3: ***R*-(1*H*-indol-3-yl)-methyl**R4: (*S*,*S*)-1-hydroxyethylR5: 4-*t*butyl-cyclohexyl-methyl6000 ± 3000(B)70% @ 1 mM75% @ 1 mM14,000 ± 20002718.059
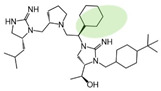
R1: *R*-isobutylR2: *S*-pyrrolidineR3: ***R*-cyclohexyl**R4: (*S*,*S*)-1-hydroxyethylR5: 4-*t*butyl-cyclohexyl-methyl70% @ 1 mM11,000 ± 4000(A)55% @ 1 mM6000 ± 1000^a^ Pyrrolidine bis-cyclic guanidine compounds were assayed at the selected receptor subtypes for agonist activity from 10^−3^ to 10^−9^ M. The results are tabulated as the mean EC_50_ values from at least three independent experiments with the reported error being the standard error of the mean (SEM). A percentage denotes the percent maximal stimulatory response (compared to NDP-MSH) observed at 1 mM concentrations, but not enough stimulation was observed to determine an EC_50_ value. The use of >1,000,000 indicates that the compound was examined but lacked agonist activity at concentrations up to 1 mM in at least two independent experiments. For partial agonist compounds (a sigmoidal dose–response curve was observed, but efficacy was less than 90% maximal NDP-MSH signal), compounds were binned into two groups: A (20–50% receptor activation) or B (51–90% receptor activation), Figure 2.


Substitution of an R3 *R*-methyl group (2718.061) resulted in a compound that could partially activate the mMC1R, mMC3R, and mMC5R at 1 mM concentrations. Elongating this functionality by three additional methylene units to an *R*-butyl group (2718.060) led to a full mMC5R agonist with 4-fold decreased potency compared to 2718.002. This compound was a partial agonist at both the mMC1R (EC_50_ = 2200 nM) and mMC3R (EC_50_ = 2200 nM), and partially activated the mMC4R. Incorporating an *R*-cyclohexyl group at the R3 position (2718.059), a group that is one methylene shorter than the equivalent R3 position in 2718.002, resulted in a full mMC5R agonist with 20-fold decreased potency relative to 2718.002. This compound was a partial mMC3R agonist (EC_50_ = 11,000 nM), and partially activated the mMC1R (70%) and mMC4R (55%) at 1 mM concentrations. Substitution of a *R*-4-hydroxy-benzyl group (2718.057) maintained mMC1R potency relative to 2718.002. This compound was a partial agonist at the mMC3R (EC_50_ = 11,000 nM) and mMC5R (EC_50_ = 6000 nM) and partially activated the mMC4R (50% at 1 mM). Similar mMC1R (EC_50_ = 1000 nM) and decreased mMC5R (EC_50_ = 8000 nM) full agonist potencies were observed when incorporating an *R*-2-methyl-naphthalene group (2718.055), which was a partial agonist at the mMC3R (EC_50_ = 7000 nM) and partially activated the mMC4R (70%). Substitution of an *R*-(1*H*-indol-3-yl)-methyl at the R3 position (2718.056) resulted in 50-fold decreased mMC5R potency relative to 2718.002, partial agonism at the mMC1R (EC_50_ = 6000 nM), and partial activation of the mMC3R and mMC4R (70% and 75%, respectively).

### 2.4. Structure–Activity Relationship Studies of 2718.003

The third compound examined, 2718.003, was previously reported to possess full mMC3R and mMC5R agonist activity (EC_50_ = 350 and 540 nM, respectively), partially stimulated the mMC1R (50–90% at 100 µM concentrations) and the mMC4R (50% activation at 10 µM concentrations), and was an antagonist at the mMC4R (pA_2_ = 5.8 with agonist MTII) [29]. In the present experiment (Table 4), full agonist activity was observed at the mMC3R (EC_50_ = 2000 nM) and mMC5R (EC_50_ = 1200 nM), as well as the mMC1R (EC_50_ = 640 nM). The 6-fold decreased mMC3R potency between the previous reported values and the present study were not seen with the other two lead compounds, though the appearance of full MC1R agonism was also observed for the 2718.001 scaffold in the present experiment. This pyrrolidine bis-cyclic guanidine was a partial agonist at the mMC4R (EC_50_ = 2500 nM) and possessed antagonist activity (pA_2_ = 5.9 with agonist NDP-MSH) at the mMC4R. Similar to the other two scaffolds, this compound replicated as an mMC3R agonist and mMC4R partial activation/antagonist.

Truncation of the R1 *R*-isopropyl group to hydrogen (2718.029) resulted in an equipotent mMC1R, mMC3R, and mMC5R agonist compared to 2718.003, with similar partial agonist (EC_50_ = 2500 nM) and antagonist (pA_2_ = 5.8) activity at the mMC4R. The truncation of the R4 *R*-propyl group to hydrogen (2718.027) also maintained mMC1R, mMC3R, and mMC5R agonist activity and mMC4R antagonist activity relative to 2718.003. Partial agonism at the mMC1R and mMC5R was observed when the R3 *R*-cyclohexyl-methyl group was replaced with hydrogen (2718.028), which also partially activated the mMC3R and mMC4R. Similar partial agonism at the mMC1R was observed when the R5 position was shortened to ethyl (2718.014) or hydrogen (271.015), with minimal-to-no partial activation of the mMC3R, mMC4R, or mMC5R. Removal of both the R4 and R5 groups (replaced with hydrogen, 2718.016) produced an mMC1R partial agonist (EC_50_ = 30,000 nM). This compound partially activated the mMC4R (25%) and mMC5R (55%), and was inactive at the mMC3R. Removal of the R1, R2, and R3 groups and amine backbone (resulting in monocyclic guanidine 2718.021) produced a full mMC5R agonist with 14-fold decreased potency relative to 2718.003. This compound partially activated the mMC1R, mMC3R, and mMC4R.
ijms-24-10145-t004_Table 4Table 4Agonist Pharmacology of Lead Molecule 2718.003 Analogs ^a^.Compound IDChemical StructureFunctionalitiesmMC1RmMC3RmMC4RmMC5REC_50_ (nM)EC_50_ (nM)EC_50_ (nM)pA_2_EC_50_ (nM)NDP-MSHAc-Ser-Tyr-Ser-Nle-Glu-His-DPhe-Arg-Trp-Gly-Lys-Pro-Val-NH_2_0.055 ± 0.0080.069 ± 0.0090.31 ± 0.04
0.10 ± 0.012718.003
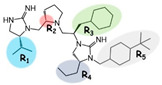
R1: *R*-isopropylR2: *S*-pyrrolidineR3: *R*-cyclohexyl-methylR4: *R*-propylR5: 4-*t*butyl-cyclohexyl-methyl640 ± 302000 ± 3002500 ± 600(A)5.9 ± 0.11200 ± 200**Functionality Truncation Analogs**2718.029
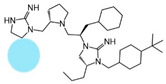
R1: **hydrogen**R2: *S*-pyrrolidineR3: *R*-cyclohexyl-methylR4: *R*-propylR5: 4-*t*butyl-cyclohexyl-methyl1100 ± 3003300 ± 1002900 ± 400(B)5.8 ± 0.11300 ± 1002718.028
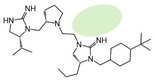
R1: *R*-isopropylR2: *S*-pyrrolidineR3: **hydrogen**R4: *R*-propylR5: 4-*t*butyl-cyclohexyl-methyl10,000 ± 6000(B)35% @ 1 mM35% @ 1 mM
19,000 ± 5000(B)2718.027
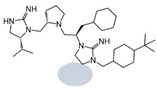
R1: *R*-isopropylR2: *S*-pyrrolidineR3: *R*-cyclohexyl-methylR4: **hydrogen**R5: 4-*t*butyl-cyclohexyl-methyl340 ± 302100 ± 20045% @ 1 mM6.2 ± 0.1510 ± 702718.014
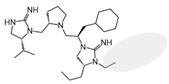
R1: *R*-isopropylR2: *S*-pyrrolidineR3: *R*-cyclohexyl-methylR4: *R*-propylR5: **ethyl**9000 ± 3000(B)20% @ 1 mM>1,000,000
50% @ 1 mM2718.015
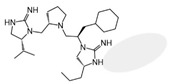
R1: *R*-isopropylR2: *S*-pyrrolidineR3: *R*-cyclohexyl-methylR4: *R*-propylR5: **hydrogen**11,000 ± 4000(B)>1,000,000>1,000,000
35% @ 1 mM2718.016
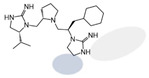
R1: *R*-isopropylR2: *S*-pyrrolidineR3: *R*-cyclohexyl-methylR4: **hydrogen**R5: **hydrogen**30,000 ± 10,000(B)>1,000,00025% @ 1 mM
55% @ 1 mM2718.021
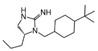
R1:R2:R3:R4: *R*-propylR5: 4-*t*butyl-cyclohexyl-methyl80% @ 100 µM50% @ 100 µM50% @ 100 µM
17,000 ± 4000**Single Residue Stereocenter Inversion Analogs**2718.053
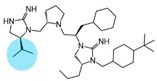
R1: ***S*-isopropyl**R2: *S*-pyrrolidineR3: *R*-cyclohexyl-methylR4: *R*-propylR5: 4-*t*butyl-cyclohexyl-methyl1000 ± 3002400 ± 20060% @ 1 mM5.9 ± 0.12400 ± 3002718.052
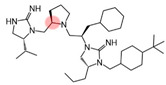
R1: *R*-isopropylR2: ***R*-pyrrolidine**R3: *R*-cyclohexyl-methylR4: *R*-propylR5: 4-*t*butyl-cyclohexyl-methyl2500 ± 3002600 ± 100(B)12,000 ± 2000(B)
2920 ± 702718.051
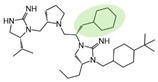
R1: *R*-isopropylR2: *S*-pyrrolidineR3: ***S*-cyclohexyl-methyl**R4: *R*-propylR5: 4-*t*butyl-cyclohexyl-methyl6000 ± 200065% @ 1 mM50,000 ± 10,000(B)
11,000 ± 30002718.050
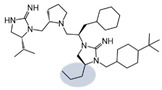
R1: *R*-isopropylR2: *S*-pyrrolidineR3: *R*-cyclohexyl-methylR4: ***S*-propyl**R5: 4-*t*butyl-cyclohexyl-methyl3200 ± 50027,000 ± 8000(B)39,000 ± 6000(B)
4200 ± 500**Quadruple Residue Stereocenter Inversion Analogs**2718.054
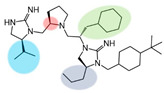
R1: ***S*-isopropyl**R2: ***R*-pyrrolidine**R3: ***S*-cyclohexyl-methyl**R4: ***S*-propyl**R5: 4-*t*butyl-cyclohexyl-methyl65% @ 1 mM55% @ 1 mM60% @ 1 mM
3500 ± 300^a^ Pyrrolidine bis-cyclic guanidine compounds were assayed at the selected receptor subtypes for agonist activity from 10^−3^ to 10^−9^ M. The results are tabulated as the mean EC_50_ or pA_2_ values from at least three independent experiments with the reported error being the standard error of the mean (SEM). A percentage denotes the percent maximal stimulatory response (compared to NDP-MSH) observed at 1 mM concentrations, but not enough stimulation was observed to determine an EC_50_ value. The use of >1,000,000 indicates that the compound was examined but lacked agonist activity at concentrations up to 1 mM in at least two independent experiments. For partial agonist compounds (a sigmoidal dose–response curve was observed, but efficacy was less than 90% maximal NDP-MSH signal), compounds were binned into two groups: A (20–50% receptor activation) or B (51–90% receptor activation), Figure 2. For compounds possessing full agonist efficacy at the MC3R and that did not fully activate the MC4R, follow-up antagonist experiments and corresponding pA_2_ values, via a Schild analysis [41], were performed at the mMC4R.


Inverting the R1 stereocenter in this scaffold to an *S*-isopropyl group (2718.053) resulted in the only stereochemical change that maintained mMC3R agonist potency relative to the corresponding lead compound. The 2718.053 ligand was an equipotent agonist (at the mMC1R, mMC3R, and mMC5R) and antagonist (mMC4R) compared to 2718.003. Changing the R2 (2718.052) and R4 (2718.050) stereochemistry resulted in equipotent mMC5R activity and decreased mMC1R (4- and 5-fold, respectively) potencies relative to 2718.003. Both changes resulted in partial agonists at the mMC3R and mMC4R. Inverting the R3 position (2718.051) decreased mMC1R and mMC5R agonist potencies 9-fold relative to 2718.003, resulted in partial agonist activity at the mMC4R, and partially activated the mMC3R. No double or triple stereochemical inversions were synthesized with this series. Inversion of the four stereochemical centers resulted in an equipotent mMC5R agonist (EC_50_ = 3500 nM) relative to 2818.003, and partially activated the mMC1R, mMC3R, and mMC4R at 1 mM concentrations.

## 3. Discussion

The present SAR study was undertaken to identify the importance of sidechain functionality and stereochemistry at five positions within the reported mMC3R agonist pyrrolidine bis-cyclic guanidine scaffold [29]. By examining three different lead templates, the overall trends for the different positions could be identified. Each of the starting compounds (2718.001, 2718.002, and 2718.003) possessed mMC3R agonist activity and were antagonists at the mMC4R. Of the additional 49 compounds synthesized, a total of 7 retained full agonist efficacy at the mMC3R (2718.024, 2718.022, 2718.026, 2718.025, 2718.029, 2718.027, and 2718.053; Figure 5). Within this set of compounds, three (2718.024, 2718.026, and 2718.029) are derived from each lead ligand with the R1 functionalities replaced with hydrogen. Another three (2718.022, 2718.025, and 2718.027) are derived from each lead ligand with the corresponding R4 functionalities replaced with hydrogen. Analyzing three distinct molecular leads within this scaffold series identifies that the R1 and R4 positions may be removed while retaining full efficacy, suggesting that the double truncation compounds may be the minimal full agonist pharmacophores at the mMC3R, albeit with decreased potency compared to the starting ligands. When comparing the scaffolds (Figure 1), the R1 and R4 are constant between 2718.001 and 2718.002 (*R*-isobutyl and (*S*,*S*)-1-hydroxyethyl, respectively), and different compared to 2718.003 (*R*-isopropyl and *R*-propyl). If these groups are removed (Figure 5 inset), the remaining core has common R2 (*S*-pyrrolidine) and R5 (4-*t*butyl-cyclohexyl-methyl) groups, and cyclic R3 groups (*R*-benzyl for 2718.001 and *R*-cyclohexyl-methyl for 2718.002 and 2718.003). The similar substructure found for the full mMC3R agonists suggests this unit may be the minimal fragment required for full MC3R activity.

Compounds that possessed full agonist efficacy at the mMC3R were also full agonists at the mMC1R and mMC5R. Within the first template (2718.001), truncation of the R1 (2718.024) or R4 (2718.022) sidechains resulted in compounds possessing nanomolar EC_50_ values at the mMC1R and mMC5R. Similar agonist potencies at the mMC1R and mMC5R were also observed for this template when the R1 (2718.033), R2 (2718.032), and R1/R2 (2718.040) positions were inverted. Changing the R2/R3 (2718.038) and R2/R4 (2718.036) stereocenters resulted in nanomolar mMC1R agonist compounds. A similar pattern was observed for truncation and stereochemical inversions in the second template (2718.002). Truncation of the R1 (2718.026) and R4 (2718.025) positions or inverting the R1 (2718.048) and R2 (2718.047) sidechains resulted in nanomolar agonist potencies at the mMC1R and mMC5R. From the third template compound, truncation of the R4 position (2718.027) was the only in this series that possessed nanomolar agonist potency at the mMC1R and mMC5R. In comparison, no compounds were observed to possess full agonist efficacy at the mMC4R. Overall, the pyrrolidine bicyclic guanidine templates examined herein appear to possess some agonist selectivity for the mMC3R over the mMC4R. Additional substitutions that decreased mMC3R efficacy and/or potency were found to maintain mMC1R and mMC5R activity, suggesting potential substitution sites that may be exploited to generate novel compounds selective for the mMC1R and/or mMC5R.

Previously, a mixture-based positional scan was used to identify the three lead mMC3R agonist compounds that were the basis of the parallel SAR studies that were the focus of the current experiments [29]. Several mixtures not initially selected for deconvolution have individual compounds represented in the current experiments, permitting a retrospective analysis on the deconvolution process and the selection criteria used in the initial deconvolution. The initial deconvolution was based upon agonist potency at the mMC3R and relative selectivity for the mMC3R over the mMC4R [29]. Substitution of hydrogen at the first position (R1 Gly) at the mixture level resulted in minimal agonist activity and no selectivity over the mMC4R [29]. Two of the individual compounds with hydrogen at the R1 position (2718.024 and 2718.026) were 10-fold less potent at the mMC3R than the respective lead compounds, supporting the mixture-based screening results. In the mixture screen, the mixture defined with hydrogen at the R4 position (R3 in the previously published table) was the fifth most active mixture and possessed some selectivity for the mMC3R over the mMC4R [29]. The individual compounds with hydrogen at this position were equipotent (2718.022 and 2718.027) or possessed 4-fold decreased potency (2718.025) at the mMC3R compared to their respective leads. The minimal loss of potency could be relatively observed in the functional activity at the mixture level. Inverting the chirality of the incorporated amino acids at the R1 [DLeu (*R*-isobutyl) and DVal (*R*-isopropyl) to Leu (*S*-isobutyl) and Val (*S*-isopropyl)], R3 [DPhe (*R*-benzyl) to Phe (*S*-benzyl)], and R4 [DThr ((*S*,*S*)-1-hydoxyethyl) and DNva (*R*-propyl) to Thr ((*R*,*R*)-1-hydroxyethyl) and Nva (*S*-propyl)] decreased activities in the corresponding mixtures [29]. Therefore, the general decreased mMC3R activities may not be surprising in the current experiments from the stereochemical inversions. Although the purpose of the current set of experiments was to examine truncation and chirality changes to known mMC3R lead compounds, the mixture-based positional scan previously performed generally predicted the observed activity trends at the different substitution positions.

While these experiments were carried out to identify the common core scaffold requirements for MC3R agonism, two smaller fragments were observed to fully activate the mMC5R (Figure 6). Compounds 2718.020 and 2718.021 were constructed by truncation of the R1, R2, and R3 positions, resulting in fragments possessing molecular weights under 300 Da that were micromolar potent mMC5R agonists (Figure 6). The core of these two compounds is a 5-member heterocyclic guanidine group. The previously reported MC1R/MC5R antagonist JNJ-10229570, possessing IC_50_ values in the 200–300 nM range [43], has a 5-member heterocyclic amino-thiadiazole core. The presence of the heterocyclic core in these compounds suggests that this motif may be important in developing MC5R agonist ligands and may offer guidance in substitutions that result in more potent compounds. 

## 4. Materials and Methods

### 4.1. Reagents

Dichloromethane (DCM), dimethylformamide (DMF), and methanol (MeOH) were purchased from Fisher (Hampton, NH, USA). 1-Hydroxybenzotriazole hydrate (HOBt), diisopropylcarbodiimide (DIC), diisopropylethylamine (DIEA), amino acids (unless otherwise noted), and resin were purchased from Chem-Impex (Wood Dale, IL, USA). Boc-Pro-OH was purchased from EMD (La Grange, IL, USA). Boc-DTrp-OH was purchased from Advanced ChemTech (Louisville, KY, USA). 4-*Tert*-butyl-cyclohexanecarboxylic acid was purchased from TCI (Portland, OR, USA). Trifluoroacetic acid (TFA) was purchased from Alfa Aesar (Haverhill, MA, USA). Borane-tetrahydrofuran was purchased from Acros Organics (Branchburg, NJ, USA). Piperidine and acetic anhydride were purchased from Sigma-Aldrich. Cyanogen bromide was purchased from ThermoScientific (Waltham, MA, USA).

### 4.2. Compound Synthesis

The compounds were synthesized (Figure 1) utilizing the “tea-bag” method [35]. The *p*-methylbenzhydrylamine (MBHA) resin was sealed in a mesh “tea-bag”, washed with dichloromethane (DCM) (3 × 1 min), neutralized with 5% diisopropylethylamine (DIEA) in DCM (3 × 2 min), and then swelled with additional DCM washes (3 × 1 min). Boc-amino acids R1 (6 eq) were coupled using a standard coupling protocol with *N*,*N*′-diisopropylcarbodiimide (DIC) (6 eq) and hydroxybenzotriazole (HOBt) (6 eq) in dimethylformamide (0.1 M) for 2 h. Following DMF (3 × 1 min) and DCM (3 × 1 min) washes, the Boc-protecting group was removed with 55% trifluoroacetic acid (TFA) in DCM (1 × 30 min) and the resin was washed with DCM (2 × 1 min) and isopropyl alcohol (IPA) (2 × 1 min). The resin was washed and neutralized with the same protocol (DCM 3 × 1 min, 5% DIEA/DCM 3 × 2 min, DCM 3 × 1 min). The amide bond coupling, Boc-protecting group removal, and neutralization steps with identical equivalents were repeated for the remaining Boc-Proline-OH R2, Boc-amino acid R3, and Boc-amino acid R4. Following the Boc removal and neutralization from the R4 amino acid, the carboxylic acid R5 was coupled with the same protocol but with an increase in equivalents of the carboxylic acid (10 eq), DIC (10 eq), and HOBt (10 eq). The amide bonds were reduced without the racemization of the side chains in the presence of 1.0 M borane (BH_3_) in tetrahydrofuran (THF) (40 eq per amide bond) using anhydrous conditions and heated to 65 °C for 72 h. The solution was slowly quenched with methanol and removed. The bags were washed with methanol (5 × 1 min) and subsequently treated with piperidine at 65 °C for 24 h. The bags were then rinsed with three cycles each of two washes of DMF followed by two washes of DCM. After neutralization of the bags, the cyclization to the bis-cyclic guanidine moieties was performed on solid support with 0.1 M cyanogen bromide (5 eq) in anhydrous DCM at room temperature for 3 h. The bags were rinsed with DMF and DCM, and then the desired compounds were removed from the solid support with HF in the presence of anisole in an ice bath at 0 °C for 1.5 h. Excess HF was removed with N_2_ gas and the product was extracted from the reaction vessel with 95% acetic acid in water, frozen, and lyophilized. 

### 4.3. Compound Purification

Individual compounds were purified using preparative HPLC with a dual pump Shimadzu LC-20AB system equipped with a Luna C18 preparative column (21.5 × 150 mm, 5 micron) at λ = 214 nm, with a mobile phase of (A) H_2_O (+0.1% formic acid)/(B) acetonitrile (+0.1% formic acid) at a flow rate of 15 mL/min; gradients varied by compound based on hydrophobicity. The compounds were lyophilized three additional times following purification. The purities of synthesized compounds were determined by LC/MS analysis on a Shimadzu LCMS-2020 instrument with ESI Mass Spec and SPD-20A Liquid Chromatograph equipped with a Luna C18 column (50 mm × 4.6 mm, 5 micron) with a mobile phase of (A) H_2_O (+0.1% formic acid)/(B) ACN (+0.1% formic acid) (5–95% over 6 min with a 4 min rinse). ^1^H NMR spectra were recorded in DMSO-d_6_ on a Bruker Ascend 400 MHz spectrometer at 400.14 MHz.

### 4.4. AlphaScreen Assay

The purified compounds were dissolved in DMSO at a stock concentration of 10^−1^ M and assayed using HEK293 cells stably expressing the mouse MC1R, MC3R, MC4R, and MC5R using the AlphaScreen cAMP bioassay (PerkinElmer, Waltham, MA, USA) according to the manufacturer’s instructions and as previously described [37,38,39].

Briefly, cells 70−90% confluent were dislodged with Versene (Gibco, Waltham, MA, USA) at 37 °C and 10,000 cells/well were plated in a 384-well plate (Optiplate, PerkinElmer, Waltham, MA, USA) with 10 μL of freshly prepared stimulation buffer (1× HBSS, 5 mM HEPES, 0.5 mM IBMX, 0.1% BSA, pH = 7.4) with 0.5 μg of anti-cAMP acceptor beads per well. The cells were stimulated with the addition of 5 μL of stimulation buffer containing compound or forskolin (10^−4^ M) and incubated in the dark at room temperature for 2 h. Following stimulation, streptavidin donor beads (0.5 μg) and biotinylated cAMP (0.62 μmol) were added to the wells in a green light environment with 10 μL of lysis buffer (5 mM HEPES, 0.3% Tween-20, 0.1% BSA, pH = 7.4) and the plates were incubated in the dark at room temperature for an additional 2 h. Plates were read on a Enspire (PerkinElmer, Waltham, MA, USA) Alpha plate reader using a pre-normalized assay protocol (set by the manufacturer).

### 4.5. Data Analysis

The pA_2_ and EC_50_ values represent the mean of at least three independent experiments performed in duplicate replicates. The compounds that were not active in two independent agonist experiments at the concentrations assayed (>1 mM for agonist assays) were not furthered examined. The pA_2_ and EC_50_ estimates and associated standard errors (SEM) were determined by fitting the data to a nonlinear least-squares analysis using the PRISM program (version 4.0, GraphPad Inc., San Diego, CA, USA). The compounds were assayed as formic acid salts. 

## 5. Conclusions

The present experiments examined parallel structural modifications of three previously reported pyrrolidine bis-cyclic guanidine templates that were full MC3R agonists. Truncation of the R1 or R4 positions in all compounds resulted in full MC3R efficacy, implying that the R1 and R4 positions may be removed while retaining activity. Truncation of the R2, R3, and R5 positions resulted in loss of MC3R efficacy, indicating that the minimal MC3R pharmacophore requires such substitutions in this scaffold. Fragments with molecular weights under 300 Da were also identified as possessing full MC5R agonist efficacy. These studies identified minimal MC3R and MC5R pharmacophores based upon cyclic guanidine scaffolds, which may be useful in developing potent, selective ligands that clarify the roles of these receptors in vivo and as therapeutic leads to treat a variety of conditions caused by dysregulation of the melanocortin receptors.

## Data Availability

The authors are happy work with individuals who would like to view raw data files. Please contact the corresponding author at: chaskell@umn.edu.

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
