# Peer review of "The Parallel Structure–Activity Relationship Screening of Three Compounds Identifies the Common Agonist Pharmacophore of Pyrrolidine Bis-Cyclic Guanidine Melanocortin-3 Receptor (MC3R) Small-Molecule Ligands"

_ijms, 2023, doi:10.3390/ijms241210145_

Round 1

Reviewer 1 Report

The manuscript reports the structural-activity relationship of a kind of MC3R small molecule inhibitors. By comparing derivatives of the inhibitors, they have a better understanding of SAR of MC3R inhibitors, which is critical for developing potent MC3R inhibitors.

I have the following suggestions and questions based on the manuscript:

1. The manuscript figures contain many IC50 curves of different compounds but do not show any repeat or standard deviation of repeats in these curves. Please include the SD or SE in these curves, and add a statement of the number of repeats for each data point in the figure legends. 

2. Besides the inhibition to MC3R, the author also tested the inhibition of the compounds to MC1,4/5R. However, I did not find much analysis about the selectivity of the inhibitors to these receptors. How is the modification on R1-R5 related to the selectivity of the compounds? 

3. Please provide the raw data of all inhibition curves in suppelmentary data. 

Author Response

We greatly appreciate the reviewers time and efforts to improve this manuscript.

Reviewer 2 Report

The manuscript contains detailed studies of interaction of a library of peptide derived small molecules with 4 melanocortin receptors. The studies were properly designed and allowed to find some relationship and consequently select the most active hit and the possible candidate for key motif (smallest active fragment).

The data are properly designed and there are only few minor questions could be addressed:

1)     Authors did not provide enough information to prove that the any racemisation did not occurs during the peptide bond formation, and especially during the reduction (the single HPLC peak may indicates both: no racemisation, and not optimal conditions of separation) the source NMR spectra were not provided.

2)     The tables contained in head the peptide sequence but there is no explanation for it provided.

3)     Since HOBt is available as both anhydrous and hydrate, it could be important to provide more information about the reagent used. The information about the origins of all key reagents could be also provided.

Author Response

(The authors gave the same response as above.)
